# Nonlinear Modeling Study of Aerodynamic Characteristics of an X38-like Vehicle at Strong Viscous Interaction Regions

**DOI:** 10.3390/e24060836

**Published:** 2022-06-17

**Authors:** Dingwu Jiang, Pei Wang, Jin Li, Meiliang Mao

**Affiliations:** Computational Aerodynamics Institute, China Aerodynamics Research and Development Center, Mianyang 621000, China; dwjiang@cardc.cn (D.J.); mlmao@cardc.cn (M.M.)

**Keywords:** X38-like vehicle, hypersonic, aerodynamic characteristics, viscous interaction effect, rarefied effect, modelling

## Abstract

Strong viscous interaction and multiple flow regimes exist when vehicles fly at high altitude and high Mach number conditions. The Navier–Stokes(NS) solver is no longer applicable in the above situation. Instead, the direct simulation Monte Carlo (DSMC) method or Boltzmann model equation solvers are usually needed. However, they are computationally more expensive than the NS solver. Therefore, it is of great engineering value to establish the aerodynamic prediction model of vehicles at high altitude and high Mach number conditions. In this paper, the hypersonic aerodynamic characteristics of an X38-like vehicle in typical conditions from 70 km to 110 km are simulated using the unified gas kinetic scheme (UGKS), which is applicable for all flow regimes. The contributions of pressure and viscous stress on the force coefficients are analyzed. The viscous interaction parameters, Mach number, and angle of attack are used as independent variables, and the difference between the force coefficients calculated by UGKS and the Euler solver is used as a dependent variable to establish a nonlinear viscous interaction model between them in the range of 70–110 km. The evaluation of the model is completed using the correlation coefficient and the relative orthogonal distance. The conventional viscous interaction effect and rarefied effect are both taken into account in the model. The model can be used to quickly obtain the hypersonic aerodynamic characteristics of X38-like vehicle in a wide range, which is meaningful for engineering design.

## 1. Introduction

The viscous interaction effect, which describes the mutual interaction process between the boundary layer and the outer inviscid flow, is one of the three main effects [1] on hypersonic vehicles for ground-to-flight extrapolation. Depending on the degree of feedback from the inviscid flow on the boundary layer, strong viscous interaction and weak viscous interaction can be defined.

Traditionally, a similarity parameter, χ¯=M∞3C/Re, is used to ascertain whether an interaction region is strong or weak. Re=ρeUex/μe is the conventional Reynolds number based on properties, $\rho_{e}$, $U_{e}$ and $\mu_{e}$ at the outer edge of the boundary layer: $C=\mu_{w}\rho_{w}/\left(\mu_{e}\rho_{e}\right)$. Large values of $\bar{\chi }$ correspond to the strong interaction and small values of $\bar{\chi }$ indicate a weak region. For pressure and force coefficients on simple configurations such as a flat plate or a sharp cone, a different correlation parameter, ν∞=M∞C/Re, is usually used. The study of viscous interaction correlation for force coefficients derived from the space shuttle program has identified a modified viscous interaction parameter $\nu_{\infty }^{\prime }$ [2], which has been widely used in the literature to correlate the aerodynamic characteristics obtained by different means such as wind tunnel, flight, or numerical calculation. $\nu_{\infty }^{\prime }$ is defined as ν∞′=M∞C′/ReL∞. ReL∞ is the Reynolds number based on the characteristic length of the vehicle. $C^{\prime }=\mu^{\prime }T_{\infty }/\left(\mu_{\infty }T^{\prime }\right)$. $T^{\prime }/T_{\infty }$ is the ratio of the reference temperature in the boundary layer to the incoming flow temperature.

Gong et al. [3] and Chen et al. [4] conducted numerical simulation and proved that for the OV-102 orbiter, $\nu_{\infty }^{\prime }$ is an accurate and effective correlation parameter for aerodynamic ground-to-flight extrapolation. Mao et al. [5] carried out correlative analyses for the viscous interaction effect based on the similarity solution for hypersonic boundary layers and concluded that the difference between the wall pressure on the surfaces of the effective body and the real body is proportional to the viscous interaction parameter at a high effective angle of attack. Hypersonic flow fields around a lifting body vehicle have been simulated by them to validate their conclusion. Han et al. [6] designed a gliding wave-rider vehicle and studied the effect of viscous interaction on the aerodynamic characteristics. It was found that the relationship between the difference of the pitching moment coefficient due to the viscous interaction and the viscous interaction parameter is nonlinear. The sign of the difference is opposite to that of the difference on the space shuttle-like vehicle, indicating that the region and intensity of the viscous interaction effects are configuration-dependent. Wang [7] proposed a joint correlative parameter to correlate experimental data with flight data for a lifting body vehicle. On the basis of experimental and numerical results of the lifting body, correlative results between joint correlative parameters with the axial force coefficient are improved efficiently compared with other parameters in terms of precision and accuracy. Zhang et al. [8] proposed a viscous interaction model of longitudinal aerodynamic coefficients under perfect gas conditions for a hypersonic wing-body configuration. The quantitative uncertainty of the prediction by the viscous interaction model is also presented in the form of relative orthogonal distance.

Molecular motion and collision at the microscopic level are two important mechanisms that determine the thermodynamic state of macroscopic fluids [9,10]. Two limiting states exist. One is the state in which the molecules are in equilibrium at all time and can be described macroscopically by the Euler equation, and the other is the state of free molecular flow without any collision between molecules. The motion of the molecules leads to viscosity. In general, the NS equation can be used when the deviation from the equilibrium state is not too great. The traditional numerical study of viscous interaction is based on the NS equation solver with the continuum assumption, so the viscous interaction model or the ground-to-flight extrapolation can only be used in the continuum regime. If the collisions between molecules are further reduced, the continuum assumption breaks down, and the so-called rarefied gas effect will appear. Rarefied gas dynamics methods are then needed to predict the rarefied effect. In views of the deviation from the thermodynamic equilibrium state, the viscous interaction effect and the rarefied gas effect are homologous. Both can be considered as the thermodynamic non-equilibrium effect. While for the viscous interaction effect flows deviate slightly from the equilibrium, for the rarefied gas effect flows deviate strongly from the equilibrium.

In fact, regardless of the amount of computation, most rarefied gas dynamics methods, such as DSMC [10] and the Boltzmann model equation solvers [11,12,13,14,15,16,17,18,19,20,21,22,23,24,25,26,27,28,29,30,31,32,33,34,35,36,37,38,39,40,41], can recover the NS solution in the continuum regime. Based on these methods, a viscous interaction model can be established for all flow regimes. Thus, both the traditional viscous interaction effect and the rarefied gas effect at high altitude are taken into account.

However, solving model equations in six dimensions for complex configurations at hypersonic conditions is always challenging work. Accuracy, efficiency, parallelization, robustness, memory cost, etc., are all concerns. Li [18,19,20,21] has developed a model solver called the gas-kinetic unified algorithm (GKUA). The GKUA has been validated and applied for many vehicles, such as the reusable sphere-cone satellite, the reentry spacecraft, and a complex wing-body combination shape. A total of 727 billion cells in a six-dimensional mesh and 23,800 cores on almost the largest computer systems available in China in 2015 were used in the last case [20]. Titarev [11,12,13,14,15,16] has developed an implicit parallel code, Nesvetay, in recent years. A breakthrough in Nesvetay is the adaptive velocity mesh which is almost linearly dependent on the free-stream Mach number [14]. For a M = 25 flow around the TsaGI reentry space vehicle, 18 billion six-dimensional mesh cells and only 5000 core-hours of computer time are consumed, which is state of the art. By comparison with DSMC results, Titarev also evaluated the BGK and Shakhov model equations as applied to hypersonic flows for both aerodynamics and heat transfer in [14,15,16]. Apart from the unstructured mesh technique used in Nesvetay, another efficient approach based on an adaptive octree velocity mesh is proposed by Baranger [17]. The octree mesh contains many fewer points than a traditional Cartesian mesh. In 2010, Xu [22] proposed the unified gas kinetic scheme (UGKS) method which is based on the integral solution of the model equation. The NS solution can be recovered from the UGKS in the hydrodynamic limit [23]. Good agreements between UGKS and DSMC results have also been achieved in rarefied regime [22,23,24,25,26,27]. Many advanced techniques such as the adaptive velocity method [28,29], implicit method [30,31,32,33,34,35], multigrid method [36], and memory-saving method [37] have been implemented. UGKS has been widely used in the simulation of flow fields from low speed to high speed, from continuum flow to rarefied flow [22,23,24,25,26,27,28,29,30,31,32,33,34,35,36,37,38,39,40]. For hypersonic validations and applications, Jiang [31] has conducted a UGKS simulation and verified its accuracy by comparing the pressure, stress, and heat flux distributions on an M = 25 cylinder for different regimes with DS2V results. Li [40] has conducted a kinetic blind comparative study on the aerodynamic characteristics of a complex-scaled X38-like vehicle which is the same as the one under study in the current paper. The free-stream Mach number is 8 with four different Knudsen numbers, 0.00275, 0.0275, 0.275, and 2.75. Two in-house kinetic solvers are used based on the DSMC method and UGKS method, respectively. Despite having different methods (statistical vs. deterministic) and different meshes (unstructured vs. structured), both UGKS and DSMC solvers gave similar and reasonably consistent results. The average relative errors for the lift and drag coefficients are only 0.98% and 2.01%, respectively.

Based on the above understanding and our practical experiences with UGKS in the past decade, the goal of this paper was to establish a viscous interaction model applicable to all regimes. An in-house UGKS solver was used to predict the aerodynamic characteristics of a complex X38-like configuration at high altitude (70–110 km) and high Mach number (≥10). The viscous interaction correlation method derived from the space shuttle program [2] was used for reference.

The difference between the aerodynamic characteristics obtained by UGKS and the solution of inviscid Euler equations was used as the dependent variable. A prediction model relating the difference and the viscous interaction parameter was proposed. The model was evaluated using the concepts of correlation coefficient and relative orthogonal distance. Some new cases were selected and calculated by UGKS and the prediction model to verify the accuracy of the model prediction results.

## 2. Numerical Methods

### 2.1. The Inviscid Solver

The governing equations are the three-dimensional compressible Euler equations in general curvilinear coordinates. The equations are discretized based on the finite volume method and solved by the implicit LUSGS method. See [41] for more details.

### 2.2. The Viscous Solver

The governing equation is the Shakhov model equation [42] which can be written in non-dimensional form:(1)ft+u⋅∇f=f+−fττ=μpRe∞,f+=gM+gM(1−Pr)8λ25c·qρ(λc2−52)gM=ρ(λπ)32e−λ((u−U)2+(v−V)2+(w−W)2),λ=γM∞22T,Re∞=ρ∞|U∞|Lrefμ∞

Here f is the distribution function which is a function of the space x, the particle velocity u, and time t. τ is the collision time. $g_{M}$ is the local Maxwellian distribution function. The second term in $f^{+}$ is a correction term based on the original BGK model equation in order to obtain a reasonable Prandtl number, Pr.c and q are the random velocity vector and the heat vector, respectively. μ, ρ, and p are the non-dimensional viscosity, density, and pressure, respectively. Re∞ and $M_{\infty }$ are the free stream Reynolds number and Mach number, respectively. Dimensional free stream density $\rho_{\infty }$, velocity modulus $\left|U_{\infty }\right|$, temperature $T_{\infty }$, and viscosity $\mu_{\infty }$ are used to obtain the non-dimensional macroscopic quantities in the following way
(2)$$p=\frac{p^{*}}{\rho_{\infty }U_{\infty }^{2}},\rho =\frac{\rho^{*}}{\rho_{\infty }},\mu =\frac{\mu^{*}}{\mu_{\infty }},T=\frac{T^{*}}{T_{\infty }},U=\frac{U^{*}}{\left|U_{\infty }\right|},V=\frac{V^{*}}{\left|U_{\infty }\right|},W=\frac{W^{*}}{\left|U_{\infty }\right|},q=\frac{q^{*}}{\rho_{\infty }U_{\infty }^{3}}$$
The superscript ‘${}^{*}$’ denotes dimensional quantities. The power-law intermolecular interaction $\mu =T^{\omega }$ is assumed. The total length of the vehicle $L_{ref}$ is used as scale of length. $t_{\infty }=L_{ref}/\left|U_{\infty }\right|$ is the scale of temporal variable. $\rho_{\infty }/{\left|U_{\infty }\right|}^{3}$ is used to obtain non-dimensional distribution function f.

Unless explicitly specified, all variables in the following are non-dimensional.

The relations between the macroscopic conserved quantities Q, the stress P, the heat q and the distribution function are
(3)$$Q=\int f\psi d\Xi \;\;\psi =(1,u,\frac{1}{2}u^{2})$$
(4)$$P=\int ccfd\Xi \;\;q=\int \frac{1}{2}c\cdot c^{2}fd\Xi$$
where ψ is the vector of moments and dΞ=dudvdw is the volume element in the phase space.

In UGKS, at the cell interface (*i* + 1/2,*j*,*k*) an integral solution of the Shakhov model in the following form is used to construct the solution:(5)$$\begin{array}{ll} f_{i+1/2,j,k,l,m,n}= & \frac{1}{\tau }\int_{0}^{t}f^{+}\left(x_{i+1/2}-u_{l}(t-t^{\prime }),t^{\prime },u_{l,}v_{m},w_{n}\right)e^{-\left(t-t^{\prime }\right)/\tau }dt^{\prime }\\ & +e^{-t/\tau }f_{0}\left(x-u_{l}t,0,u_{l,}v_{m},w_{n}\right) \end{array}$$
where $f^{+}=g+g^{+}$ will be approximated separately. The subscripts *i*,*j*,*k* and *l*,*m*,*n* denote the indexes in three structured physical mesh directions and three Cartesian velocity mesh directions, respectively. $x^{\prime }=x_{i+1/2}-u_{l}(t-t^{\prime })$ is the particle trajectory and $f_{0}$ is the initial gas distribution function at the beginning of each time step around the cell interface $x_{i+1/2}$ at particle velocity $u=\left(u_{l},v_{m},w_{n}\right)$.

As the distribution function inside each control volume is known at the beginning of each time step. $f_{0}$ can be obtained using TVD reconstruction.
(6)$$f_{0,l,m,n}=\{\begin{array}{c} f_{i+1/2,j,k,l,m,n}^{L}+\sigma_{i,j,k,l,m,n}x\;x\leq 0\\ f_{i+1/2,j,k,l,m,n}^{R}+\sigma_{i+1,j,k,l,m,n}x\;x>0 \end{array}$$
where a nonlinear limiter is used to reconstruct $f_{i+1/2,j,k,l,m,n}^{L}$,$f_{i+1/2,j,k,l,m,n}^{R}$ and the corresponding slopes $\sigma_{i,j,k,l,m,n}$,$\sigma_{i+1,j,k,l,m,n}$.

The equilibrium state g around the cell interface $x_{i+1/2}$ can be expanded with two slopes
(7)$$g=g_{0}\left[1+\left(1-H\left[x\right]\right){\bar{a}}^{L}x+H\left[x\right]{\bar{a}}^{R}x+\bar{A}t\right]$$
where H[x] is the Heaviside function. $g_{0}$ is a local Maxwellian distribution located at the cell interface. It can be determined by the corresponding macroscopic flow variables. ${\bar{a}}^{L}$, ${\bar{a}}^{R}$, and $\bar{A}$ are related to the derivatives of a Maxwellian distribution in space and time. For details to obtain $g_{0}$, ${\bar{a}}^{L}$, ${\bar{a}}^{R}$ and $\bar{A}$, see [22,25,26].

With the determination of equilibrium state and the heat flux at the cell interface, the additional term $g^{+}$ in the Shakhov model can be determined.

Substituting Equations (6) and (7) into Equation (5), the gas distribution function at the cell interface with particle velocity $\left(u_{l},v_{m},w_{n}\right)$ can be expressed as
(8)$$\begin{array}{l} f_{i+1/2,j,k,l,m,n}\left(x_{j+1/2},y_{j},z_{k},t,u_{l},v_{m},w_{n}\right)\\ =\left(1-e^{-t/\tau }\right)\left(g_{0}+g^{+}\right)\\ +\left(\tau \left(-1+e^{-t/\tau }\right)+te^{-t/\tau }\right)\left({\bar{a}}^{L}H\left[u_{l}\right]+{\bar{a}}^{R}\left(1-H\left[u_{l}\right]\right)\right)u_{l}g_{0}\\ +\tau \left(t/\tau -1+e^{-t/\tau }\right)\bar{A}g_{0}\\ +e^{-t/\tau }\left(f_{i+1/2,j,k,l,m,n}^{L}H\left[u_{l}\right]+f_{i+1/2,j,k,l,m,n}^{R}\left(1-H\left[u_{l}\right]\right)\right)\\ -te^{-t/\tau }\left(\sigma_{i,j,k,l,m,n}u_{l}H\left[u_{l}\right]+\sigma_{i+1,j,k,l,m,n}u_{l}\left(1-H\left[u_{l}\right]\right)\right) \end{array}$$
From the cell interface distribution function we can obtain the distribution function flux and macroscopic flux. We will update the macroscopic variables first with the macroscopic fluxes. Subsequently, we can immediately obtain the local Maxwellian $g_{M}^{\zeta +1}$ and the additional term $f^{+,\zeta +1}$ at ζ+1 time step inside each cell. Therefore, based on Equation (1) the update of distribution function in UGKS becomes
(9)$$\begin{array}{ll} \Delta f_{i,j,k,l,m,n}= & f_{i,j,k,l,m,n}^{\varsigma +1}-f_{i,j,k,l,m,n}^{\varsigma }=-\int_{0}^{\Delta t}\left[\begin{array}{l} {\left(ff\cdot S\right)}_{i+1/2,j,k}-{\left(ff\cdot S\right)}_{i-1/2,j,k}\\ +{\left(ff\cdot S\right)}_{i,j+1/2,k}-{\left(ff\cdot S\right)}_{i,j-1/2,k}\\ +{\left(ff\cdot S\right)}_{i,j,k+1/2}-{\left(ff\cdot S\right)}_{i,j,k-1/2} \end{array}\right]dt\\ & +\frac{\Delta t}{2}\left(\frac{f_{i,j,k,l,m,n}^{+}{}^{\varsigma +1}-f_{i,j,k,l,m,n}^{\varsigma +1}}{\tau_{i,j,k}^{\varsigma +1}}+\frac{f_{i,j,k,l,m,n}^{+}{}^{\varsigma }-f_{i,j,k,l,m,n}^{\varsigma }}{\tau_{i,j,k}^{\varsigma }}\right) \end{array}$$
where ff is the distribution function flux across the interface and S is the interface area. The trapezoidal rule has been used for time integration of the collision time.

Equation (9) can be rearranged as
(10)$$f_{i,j,k,l,m,n}^{\varsigma +1}={\left(1+\frac{\Delta t}{2\tau_{i,j,k}^{\varsigma +1}}\right)}^{-1}\left\{\begin{array}{l} -\int_{0}^{\Delta t}\left[\begin{array}{l} {\left(ff\cdot S\right)}_{i+1/2,j,k}-{\left(ff\cdot S\right)}_{i-1/2,j,k}\\ +{\left(ff\cdot S\right)}_{i,j+1/2,k}-{\left(ff\cdot S\right)}_{i,j-1/2,k}\\ +{\left(ff\cdot S\right)}_{i,j,k+1/2}-{\left(ff\cdot S\right)}_{i,j,k-1/2} \end{array}\right]dt\\ +\frac{\Delta t}{2}\left(\frac{f_{i,j,k,l,m,n}^{+}{}^{\varsigma +1}}{\tau_{i,j,k}^{\varsigma +1}}+\frac{f_{i,j,k,l,m,n}^{+}{}^{\varsigma }-f_{i,j,k,l,m,n}^{\varsigma }}{\tau_{i,j,k}^{\varsigma }}\right) \end{array}\right\}$$
This is the original explicit UGKS in [22,25].

To accelerate the convergence for steady flow, the authors of [34] introduced the implicit discrete ordinate method for an unstructured physical mesh [12,13] into UGKS. A brief introduction is given below.

Rewriting Equation (1) for f with a particle velocity $u=\left(u_{l},v_{m},w_{n}\right)$ in a physical space cell (i,j,k)
(11)$$\frac{\partial f_{i,j,k,l,m,n}}{\partial t}+u_{l}\frac{\partial f_{i,j,k,l,m,n}}{\partial x}+v_{m}\frac{\partial f_{i,j,k,l,m,n}}{\partial y}+w_{n}\frac{\partial f_{i,j,k,l,m,n}}{\partial z}=\frac{\left(f_{i,j,k,l,m,n}^{+}-f_{i,j,k,l,m,n}\right)}{\tau }$$
Treating the loss term of collision integral semi-implicitly and the gain term explicitly we can find
(12)$$\begin{array}{ll} (1+\Delta t\cdot \frac{1}{\tau^{\zeta }} & +\Delta t\cdot u_{l,m,n}\nabla ){\left(\Delta f\right)}_{i,j,k,l,m,n}=\Delta t\cdot R_{i,j,k,l,m,n}^{\zeta }\\ R_{i,j,k,l,m,n}^{\zeta } & =-u_{l}\frac{\partial f_{i,j,k,l,m,n}^{\zeta }}{\partial x}-v_{m}\frac{\partial f_{i,j,k,l,m,n}^{\zeta }}{\partial y}-w_{n}\frac{\partial f_{i,j,k,l,m,n}^{\zeta }}{\partial z}+\frac{1}{\tau^{\zeta }}\left(f^{+}-f\right)\\ & =-R^{\prime }+\frac{1}{\tau^{\zeta }}\left(f^{+}-f\right) \end{array}$$
where $R^{\prime }$ is the net cell flux averaged over the evolution time step, which can be expressed as
(13)$$R^{\prime }=\frac{1}{\Delta \tilde{t}}\int_{0}^{\Delta \tilde{t}}\left[\begin{array}{l} {\left(ff\cdot S\right)}_{i+1/2,j,k}-{\left(ff\cdot S\right)}_{i-1/2,j,k}\\ +{\left(ff\cdot S\right)}_{i,j+1/2,k}-{\left(ff\cdot S\right)}_{i,j-1/2,k}\\ +{\left(ff\cdot S\right)}_{i,j,k+1/2}-{\left(ff\cdot S\right)}_{i,j,k-1/2} \end{array}\right]dt$$
The evolution time step $\Delta \tilde{t}$ is determined by the following
(14)$$\Delta \tilde{t}\leq \frac{\Delta t_{\min}}{CFL}$$
where $\Delta t_{\min}$ is the minimum marching time step determined by the stability condition. CFL is the CFL number.

Equation (12) can be further written as
(15)$$\begin{array}{l} \left(1+\Delta t\cdot \frac{1}{\tau^{\zeta }}\right){\left(\Delta f\right)}_{i,j,k,l,m,n}\\ +\frac{\Delta t}{\left|V_{i,j,k}\right|}\sum_{ii=1}^{6}(u_{l,m,n}\cdot n_{ii})\cdot \left|S_{i,j,k,ii}\right|\cdot FF\left({\left(\Delta f\right)}_{i,j,k,l,m,n},{\left(\Delta f\right)}_{i1,j1,k1,,l,m,n}\right)\\ =\Delta t\cdot R_{i,j,k,l,m,n}^{\zeta } \end{array}$$
where the subscript ii indicates the six faces of the physical cell (i,j,k). $S_{i,j,k,ii}$ is the area of the iith face. $V_{i,j,k}$ is the cell volume. The subscript (i1,j1,k1) indicates the cell which shares the iith face with cell (i,j,k). $n_{ii}$ is the outer normal vector of the iith face.
(16)$$\begin{array}{l} FF\left({\left(\Delta f\right)}_{i,j,k,l,m,n},{\left(\Delta f\right)}_{i1,j1,k1,l,m,n}\right)\\ =\frac{1}{2}\left[{\left(\Delta f\right)}_{i,j,k,l,m,n}+{\left(\Delta f\right)}_{i1,j1,k1,l,m,n}\right]\\ +\frac{1}{2}sign\left(u_{l,m,n}\cdot n_{ii}\right)\left[{\left(\Delta f\right)}_{i,j,k,l,m,n}-{\left(\Delta f\right)}_{i1,j1,k1,l,m,n}\right] \end{array}$$
Substituting Equation (16) into Equation (15), we can obtain
(17)$$\begin{array}{l} \left(1+\Delta t\cdot \frac{1}{\tau^{\zeta }}\right){\left(\Delta f\right)}_{i,j,k,l,m,n}\\ +\frac{\Delta t}{\left|V_{i,j,k}\right|}\sum_{ii=1}^{6}\left(u_{l,m,n}\cdot n_{ii}\right)\cdot \left|S_{i,j,k,ii}\right|\left[\frac{1}{2}\left(1+sign\left(u_{l,m,n}\cdot n_{ii}\right)\right)\cdot {\left(\Delta f\right)}_{i,j,k,l,m,n}\right]\\ +\frac{\Delta t}{\left|V_{i,j,k}\right|}\sum_{ii=1}^{6}\left(u_{l,m,n}\cdot n_{ii}\right)\cdot \left|S_{i,j,k,ii}\right|\left[\frac{1}{2}\left(1-sign\left(u_{l,m,n}\cdot n_{ii}\right)\right)\cdot {\left(\Delta f\right)}_{i1,j1,k1,l,m,n}\right]\\ =\Delta t\cdot R_{i,j,k,l,m,n}^{\zeta } \end{array}$$
After a simple deformation, it can be written as
(18)$$\begin{array}{l} \left[1+\Delta t\cdot \frac{1}{\tau^{\zeta }}+\Delta t\cdot b_{i,j,k,l,m,n}\right]{\left(\Delta f\right)}_{i,j,k,l,m,n}\\ +\sum_{ii=1}^{6}\Delta t\cdot c_{i,j,k,l,m,n}\cdot {\left(\Delta f\right)}_{i1,j1,k1,l,m,n}=\Delta t\cdot R_{i,j,k,l,m,n}^{\zeta }\\ b_{i,j,k,l,m,n}=\sum_{ii=1}^{6}\left(u_{l,m,n}\cdot n_{ii}\right)\cdot \left(1+sign\left(u_{l,m,n}\cdot n_{ii}\right)\right)\frac{\left|S_{i,j,k,ii}\right|}{2\left|V_{i,j,k}\right|}\\ c_{i,j,k,l,m,n}=(u_{l,m,n}\cdot n_{ii})\cdot (1-sign(u_{l,m,n}\cdot n_{ii}))\frac{\left|S_{i,j,k,ii}\right|}{2\left|V_{i,j,k}\right|} \end{array}$$
Continuing to deform
(19)$$\begin{array}{l} {\left(\Delta f\right)}_{i,j,k,l,m,n}+\sum_{ii=1}^{6}\Delta t\cdot z_{i,j,k,l,m,n}\cdot {\left(\Delta f\right)}_{i1,j1,k1,l,m,n}=\frac{\Delta t}{\chi_{i,j,k,l,m,n}}\cdot R_{i,j,k,l,m,n}^{\zeta }\\ \chi_{i,j,k,l,m,n}=1+\Delta t\cdot \frac{1}{\tau^{\zeta }}+\Delta t\cdot b_{i,j,k,l,m,n}\\ z_{i,j,k,l,m,n}=\frac{c_{i,j,k,l,m,n}}{\chi_{i,j,k,l,m,n}} \end{array}$$
Writing in matrix form
(20)$$\left(I+\Delta t\cdot Z_{l,m,n}\right)\cdot {\left(\Delta f\right)}_{l,m,n}=\Delta t\cdot {Χ}_{l,m,n}^{-1}\cdot R_{l,m,n}^{\zeta }\;{\left(\Delta f\right)}_{l,m,n}=\left(\begin{array}{l} {\left(\Delta f\right)}_{1,1,1,l,m,n}\\ {\left(\Delta f\right)}_{2,1,1,l,m,n}\\ \cdots \\ {\left(\Delta f\right)}_{NI-1,NJ-1,NK-1,l,m,n} \end{array}\right)R_{l,m,n}^{\zeta }=\left(\begin{array}{l} R_{1,1,1,l,m,n}^{\zeta }\\ R_{2,1,1,l,m,n}^{\zeta }\\ \cdots \\ R_{NI-1,NJ-1,NK-1,l,m,n}^{\zeta } \end{array}\right)\;{Χ}_{l,m,n}=\left(\begin{array}{llll} \chi_{1,1,1,l,m,n} & 0 & \cdots & 0\\ 0 & \chi_{2,1,1,l,m,n} & \cdots & 0\\ 0 & 0 & \cdots & 0\\ 0 & 0 & \cdots & \chi_{NI-1,NJ-1,NK-1,l,m,n} \end{array}\right)$$
where $\left(I+\Delta t\cdot {Ζ}_{l,m,n}\right)$ is a seven-diagonal matrix. *NI*, *NJ*, and *NK* are the total points in the *i*, *j*, and *k* directions of a block in the structured physical mesh, respectively. Applying the LU decomposition yields
(21)$$I+\Delta t\cdot Z_{l,m,n}=L_{l,m,n}\cdot U_{l,m,n}+○(\Delta t^{2})$$
$L_{l,m,n}$, $U_{l,m,n}$ are both matrices.
(22)$$\begin{array}{l} l_{pq}=\left\{ \begin{array}{l} \Delta t\cdot z_{pq}\;p<q\\ 0\;p>q \end{array}\right.\\ u_{pq}=\left\{ \begin{array}{l} 0\;p<q\\ \Delta t\cdot z_{pq}\;p>q \end{array}\right.\\ l_{pp}=u_{pp}=1 \end{array}$$
The final form of the implicit UGKS is
(23)$$L_{l,m,n}\cdot U_{l,m,n}\cdot {\left(\Delta f\right)}_{l,m,n}=\Delta t\cdot {Χ}_{l,m,n}^{-1}\cdot R_{l,m,n}^{\zeta }$$
By performing direct and backward substitutions in a structured physical mesh, ${\left(\Delta f\right)}_{i,j,k,l,m,n}$ can be found. We can then obtain the distribution function $f_{i,j,k,l,m,n}$ at time step ς+1. After that, macroscopic variables can be obtained with Equations (3) and (4).

The tests [34] on the flows over a cylinder with different free stream Mach numbers showed that the above implicit method can give the same result as the original explicit method with a properly chosen evolving time step. Meanwhile, the computational efficiency can be improved by 1~2 orders.

Due to the explicit treatment of $f_{i,j,k,l,m,n}^{+}$ in the above method, slow convergence exists in small Knudsen number cases. To further accelerate the convergence, Zhu et al. [35] proposed a macroscopic variable prediction technique to deal with $f_{i,j,k,l,m,n}^{+}$ in their implicit UGKS, which is proved to be efficient in all flow regimes.

Under the support of the National Numerical Wind Tunnel Program, an aerodynamic characteristics prediction software applicable for multiple flow regimes called NNW-UGKS [38] has been established, and the viscous flow in the current paper was simulated by this software. Decomposition both in the physical and velocity meshes is applied for MPI parallelism, which is similar to the one in [39]. The composite Newton–Cotes quadrature formula which can be used for any kinds of flow simulation including the current hypersonic or highly non-equilibrium flows, was chosen for integration.

The diffusive reflection wall boundary condition and perfect gas assumption was used.

## 3. Results and Modeling

As a demonstrator of the Crew Return Vehicle (CRV), the X38 vehicle has a number of advantages, such as relatively high lift-to-drag ratio and volumetric efficiency [43]. Although the X38 project has long been terminated, research on similar shapes still continues.

The sketch of the vehicle is shown in Figure 1. The reference length of the vehicle, $L_{ref}$, is 4.67 m.

Free-stream conditions are given in Table 1. A total number of 24 cases and 4 cases are simulated by viscous and inviscid solvers, respectively. The structured physical mesh is illustrated in Figure 2. The number of cells is 334,434 for altitudes lower than 110 km. For 110 km, the outer boundary is not large enough and additional 82,656 cells were added. The minimum distance near the wall is 1.67 mm which is nearly two times and 0.3% of the free stream mean free paths of 70 km and 110 km, respectively. The velocity mesh is 65 × 65 × 65 and 81 × 81 × 81 for M = 10 and M = 15, respectively, ranging from $-2.5\left|U_{\infty }\right|$ to $2.5\left|U_{\infty }\right|$.

To conduct a thorough mesh convergence for such a problem is almost impossible. As is shown in our previous paper [40] for a 1:16.7 scaled model, good agreements with the DSMC results can be obtained for four different free stream conditions. For the DSMC method, the cell size should be adjusted according to the free stream condition to be smaller than the local mean free path of particles. While for UGKS method, the same physical structured mesh can be used for different free stream conditions. This may be due to the coupling mechanism of the particle transport and collision in UGKS method. The cell size can be larger than the mean free path of particles.

### 3.1. Flow Field Characteristics

Figure 3 shows the pressure contour of the flow field and the velocity vector on the symmetry plane at two altitudes. For sake of clarity, the grid in the vector diagram is one out of three. The viscous boundary layer can be clearly distinguished from the figure. With the increase in altitude, the shock stand-off distance and the thickness of boundary layer increase, and the wall slip velocity, increases obviously.

Figure 4 shows the streamlines on the symmetry plane and near the body surface. No flow separation on the windward and leeward sides can be observed. At 70 km, there is a small separation at the bottom. At 100 km, no separation exists due to the smaller bottom adverse pressure gradient.

Figure 5 shows the local Knudsen number distribution on the symmetry plane and near the body surface at two altitudes.

The local Knudsen number is defined [44] as
(24)$$Kn_{GLL}=\frac{l_{mfp}}{\rho /\left|\nabla \rho \right|}$$
where $l_{mfp}$ is the local mean free path. The Knudsen number of this form has a great physical meaning. Traditionally, different flow regimes are defined according to the Knudsen numbers [10]. For continuum regime, *Kn* is smaller than 0.01. For a transitional regime, *Kn* ranges between 0.01 and 10. When *Kn* is larger than 10, the flow is considered as free-molecular. Thus, when $Kn_{GLL}$ is much less than unity the flow can be regarded as locally slightly perturbed from equilibrium which is a fundamental assumption of the NS equations. Therefore, it is an appropriate parameter to indicate the degree of non-equilibrium.

Figure 6 shows the local Knudsen number comparison along the y = 500 mm line in front of the vehicle. The local Knudsen number is large inside the bow shock which usually locates in the first peak from left, and near the wall which has been marked on the right. Even at 70 km, the local Knudsen number near the wall and inside the shock is on the order of 0.01, where the continuum assumption may break down. Thus, it is necessary to use UGKS for simulation.

Figure 7 shows the comparison of the pressure distribution on the centerlines. The pressure distribution on the windward centerline shows an increasing trend with the increase in altitude. While on the leeward centerline it increases first and then decreases with the increase in altitude. The magnitude is about an order smaller than that on the windward centerline.

Figure 8 shows the variation in the pressure change, Δp, due to viscous interaction at several typical stream-wise positions on the centerlines.

In early research on simple configurations, figures similar to Figure 8 have been frequently given and linear relationships have been obtained. For the current complex vehicle, in the range of 70~85 km and X/L = 0.1~0.5, there is a good linear relationship between the pressure change and the viscous interaction parameter on the windward side for 20 degrees angle of attack. In other areas and the whole leeward side, no good linear relationships can be seen.

### 3.2. Aerodynamic Characteristics and Viscous Interaction Modelling

Figure 9 shows the aerodynamic force coefficients computed by the Euler and UGKS solvers. For the UGKS results, the contributions of the pressure and friction are separated. With the increase in the viscous interaction parameter, the axial force and the normal force coefficients increase, and the pressure part and viscous part also increase at the same time. For the axial force, the viscous part increases rapidly as the altitude increases, from 34% at 70 km to 87% at 110 km. At 80 km and above, the viscous part exceeds the pressure part. For the normal force, the pressure part is dominant, decreasing from 95% at 70 km to 74% at 110 km, and the viscous part is relatively small.

Figure 10 shows the viscous force coefficients with the third viscous interaction parameter. Note that the viscous force coefficient is defined as the quantity due to viscous interaction [45] which is equal to the difference between UGKS and Euler solutions. Thus, it is different from the viscous part of UGKS.

At an altitude less than 100 km where $\nu_{\infty }^{\prime }$ is about 0.33 (M = 10), the viscous axial force coefficient has a weak linear relationship with the third viscous interaction parameter. The higher the altitude is, the more serious the deviation from the linear relationship is. In order to correlate the results for all ranges of calculation, it is assumed that the change in the aerodynamic coefficients due to the viscous interaction satisfies the following relationship:(25)$$\Delta C\left(\nu_{\infty }{}^{\prime },M_{\infty },\alpha \right)\approx a\left(M_{\infty },\alpha \right)+b\left(M_{\infty },\alpha \right)\nu_{\infty }{}^{\prime }+c\left(M_{\infty },\alpha \right)\nu_{\infty }{{}^{\prime }}^{2}$$
As a preliminary study, it is further assumed that
(26)$$\begin{array}{l} a\left(M_{\infty },\alpha \right)\approx a_{0}+a_{1}\left(M_{\infty }\right)+a_{2}\left(\alpha \right)\\ b\left(M_{\infty },\alpha \right)\approx b_{0}+b_{1}\left(M_{\infty }\right)+b_{2}\left(\alpha \right)\\ c\left(M_{\infty },\alpha \right)\approx c_{0}+c_{1}\left(M_{\infty }\right)+c_{2}\left(\alpha \right) \end{array}$$
According to the calculated aerodynamic force coefficients and the parameters in Equation (26), the following expression of the viscous axial force coefficient can be obtained by fitting with the least square method,
(27)$$\Delta C_{A}\left(\nu_{\infty }{}^{\prime },M_{\infty },\alpha \right)\approx a^{A}\left(M_{\infty },\alpha \right)+b^{A}\left(M_{\infty },\alpha \right)\cdot \nu_{\infty }{}^{\prime }+c^{A}\left(M_{\infty },\alpha \right)\cdot \nu_{\infty }{{}^{\prime }}^{2}\;\begin{array}{l} a^{A}=-1.26\times 10^{-2}-\frac{3.21\times 10^{-3}}{M_{\infty }^{2}}+6.14\times 10^{-4}\cdot \alpha -5.38\times 10^{-6}\cdot \alpha^{2}\\ b^{A}=1.31+\frac{2.72\times 10^{1}}{M_{\infty }^{2}}-2.23\times 10^{-4}\cdot \alpha -7.08\times 10^{-5}\cdot \alpha^{2}\\ c^{A}=-0.787-\frac{2.81\times 10^{1}}{M_{\infty }^{2}}+1.69\times 10^{-2}\cdot \alpha -1.36\times 10^{-4}\cdot \alpha^{2} \end{array}$$
From the expressions of $a^{A}$, $b^{A}$, and $c^{A}$, we can conclude that the effect of Mach number can be ignored under the condition of high Mach number. In fact, the effect of Mach number may be mainly reflected in the viscous interaction parameter.

Usually, Equation (27) is called a viscous interaction model for the axial force coefficient. Models for other viscous force coefficients can be obtained in a similar way.

Figure 11 shows the correlation curve between the viscous interaction model prediction data which is represented by suffix ‘_ model’ and the numerical simulation data which is represented by suffix ‘_ UGKS’. The data are basically distributed near the correlation line at different angles of attack and Mach numbers. It can be seen that the correlation between the data is good.

The Pearson Correlation Coefficient *r*, which is widely used in statistics, is chosen to characterize the degree of correlation between the aerodynamic prediction data and the numerical simulation data, and its expression is
(28)$$r=\frac{\sum_{i=0}^{n}\left(x_{i}-\bar{x}\right)\left(y_{i}-\bar{y}\right)}{\sqrt{\sum_{i=0}^{n}{\left(x_{i}-\bar{x}\right)}^{2}\cdot \sum_{i=0}^{n}{\left(y_{i}-\bar{y}\right)}^{2}}}$$
The closer *r* is to 1, the better agreement between the predicted values of the model and the numerical results we obtain. The Pearson correlation coefficients of axial force, normal force and pitching moment are 0.999996, 0.999973, and 0.999863, respectively. They are all very close to 1, indicating that the correlation between the predicted data and the numerical simulation data is very good.

In order to further assess the viscous interaction model, the relative orthogonal distance, $dr_{i}$, is defined to characterize the relative degree of deviation of the data from the correlation curve, as shown in the following,
(29)$$dr_{i}=\frac{d_{i}}{x_{r}}$$
The $dr_{i}$ of the viscous axial force is shown in Figure 12. The maximum fitting deviation is only 1.8%.

Finally, the accuracy of the prediction model is preliminarily evaluated. The UGKS and the viscous interaction models are used to calculate two new cases with altitudes equal to 80 km and 90 km, respectively. The angle of attack is 30 degrees with a Mach number of 15. The results and relative errors are shown in Table 2. The relative error of viscous axial force is small partially due to its large magnitude. While the error of viscous pitching moment is large due to its small magnitude compared with the viscous axial force. However, the relative error of the pitching moment itself is small. Taking the 80 km case as an example, the relative error of predicted viscous pitching moment is 9.87%. However, the pitching moments obtained by UGKS simulation and predicted by the model are −0.2158 and −0.2180, respectively, resulting in a relative error of only 1.01%.

## 4. Conclusions

Hypersonic viscous and inviscid flow fields around the X38-like vehicle are simulated by UGKS solver and Euler solver, respectively. Viscous force coefficients at different altitudes, Mach numbers, and attack angles are obtained by subtracting the two solutions and correlated by the third viscous interaction parameter. A nonlinear viscous interaction model of force coefficients is established, and some preliminary conclusions are as follows,

(1) For the X38-like vehicle, the contribution of the viscous part to the axial force coefficients increases rapidly with altitude, and reaches 87% at 110 km for the typical conditions, with Ma = 10 and AOA = 20. The contribution of the viscous part to the normal force coefficients is small, and can only reach 26% at 110 km.

(2) For complex configurations such as the current X38-like vehicle, the changes of wall pressure and aerodynamic coefficients due to viscous interaction cannot be expressed linearly with the viscous interaction parameters in the whole flow field.

(3) A viscous interaction model can be established by taking the viscous interaction parameters as the independent variables combined with the inviscid solution and the viscous solution, which is helpful to quickly obtain the aerodynamic characteristics at moderate to high altitudes and has certain application value in engineering design.

In this paper, the idea of modeling the viscous interaction based on UGKS solver is applied to the X38-like vehicle, and a satisfactory result has been achieved. The prediction model can take into account both the viscous interaction effect and rarefied gas effect. However, the Cartesian velocity mesh in our UGKS solver causes huge waste both in computation and memory. The next step is to introduce an unstructured velocity mesh into our solver to reduce the cost and give a more accurate prediction model for more complex configurations.

## Figures and Tables

**Figure 1 entropy-24-00836-f001:**
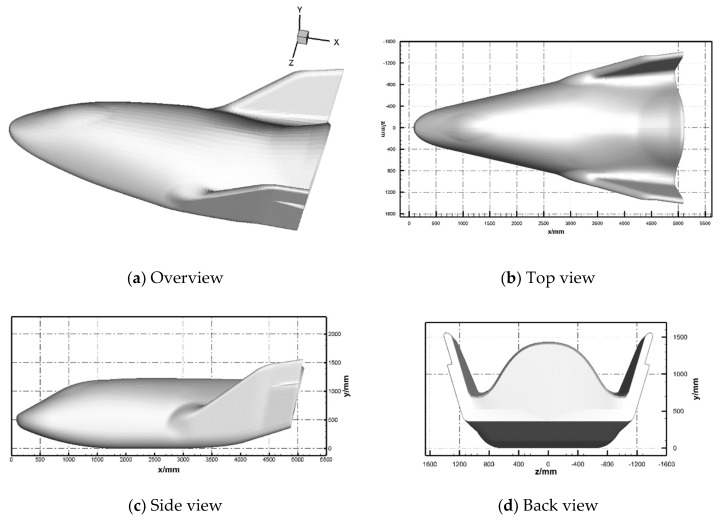
Sketch of the X38-like vehicle.

**Figure 2 entropy-24-00836-f002:**
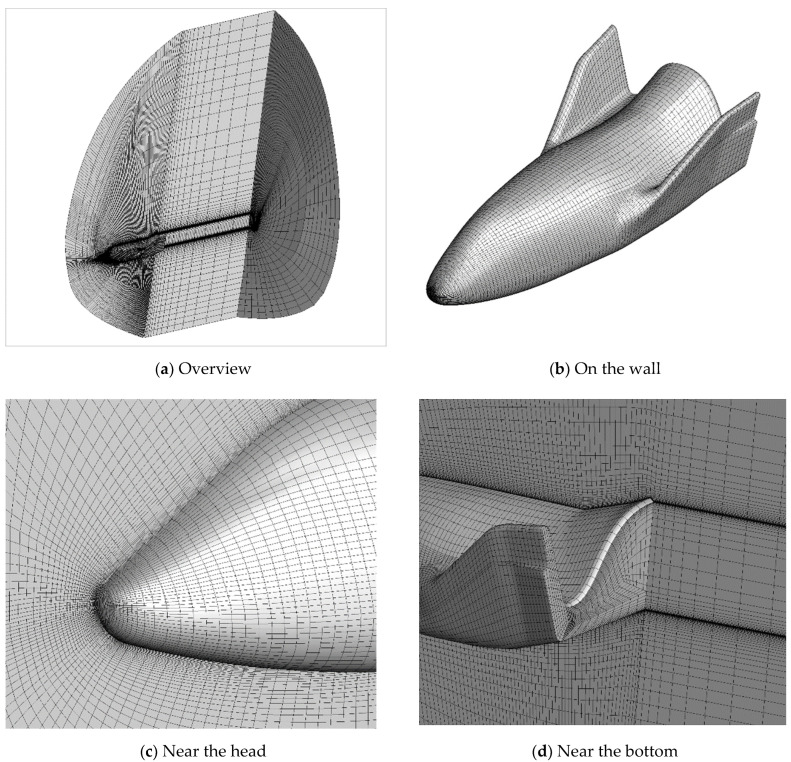
Structured multi-block physical mesh.

**Figure 3 entropy-24-00836-f003:**
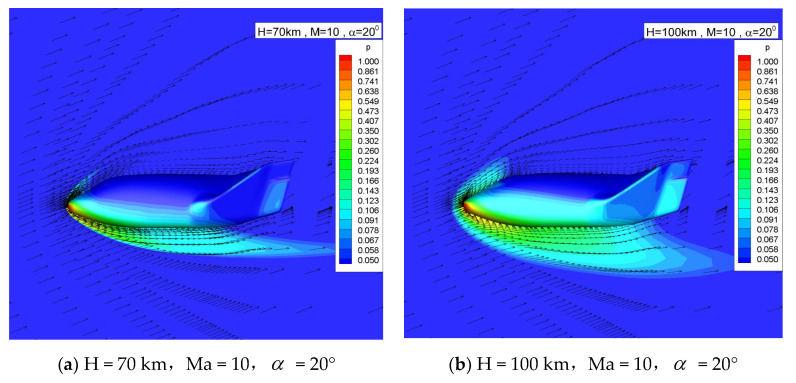
Velocity vector and pressure contour.

**Figure 4 entropy-24-00836-f004:**
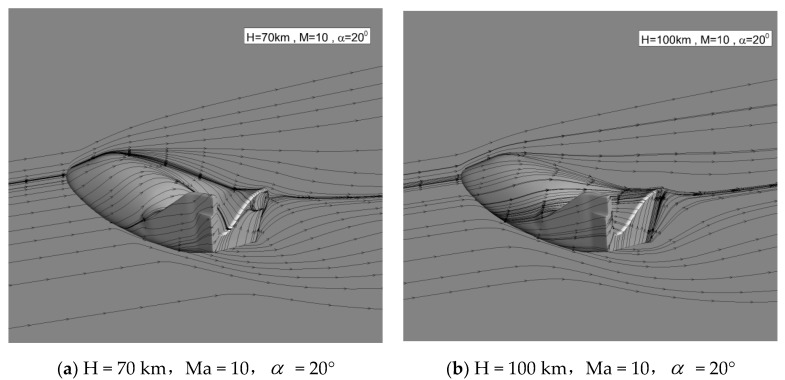
Streamlines on the symmetry plane and near the body surface.

**Figure 5 entropy-24-00836-f005:**
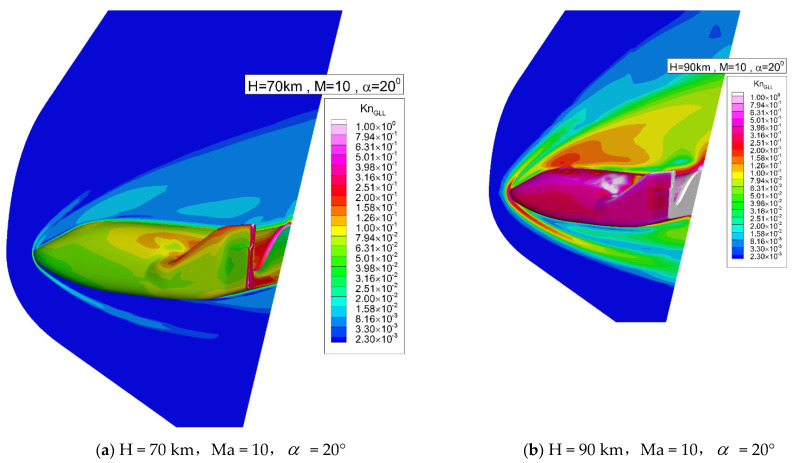
Local Knudsen number distribution on the symmetry plane and near the body surface.

**Figure 6 entropy-24-00836-f006:**
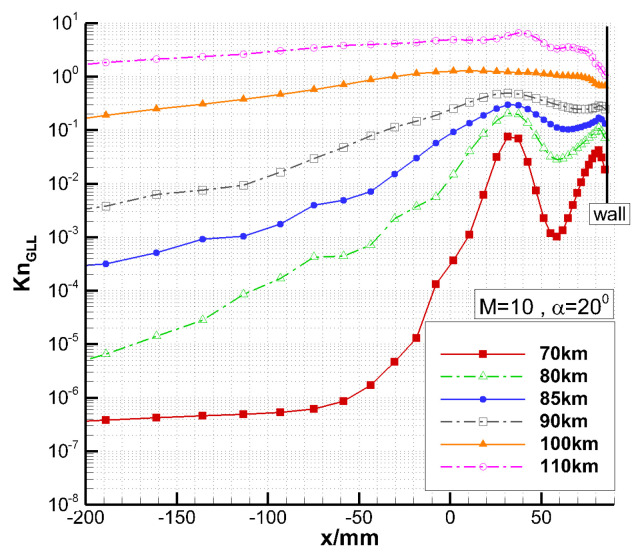
Local Knudsen number comparison along the y = 500 mm line.

**Figure 7 entropy-24-00836-f007:**
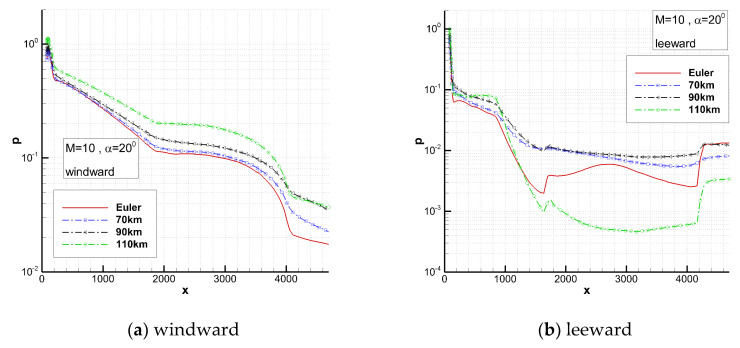
Comparison of pressure on the centerlines.

**Figure 8 entropy-24-00836-f008:**
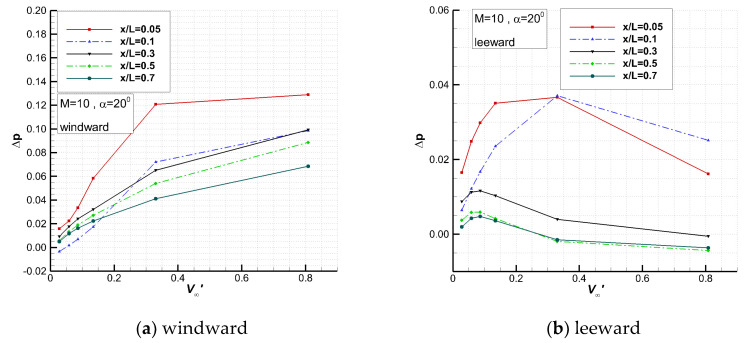
Variation in pressure change on the centerlines with viscous interaction parameter.

**Figure 9 entropy-24-00836-f009:**
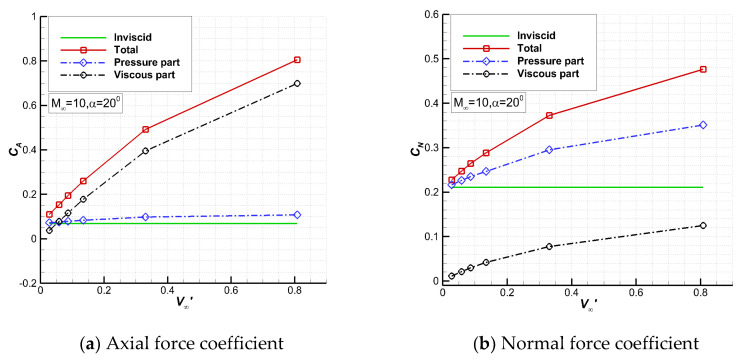
Aerodynamic force coefficients for different $\nu_{\infty }^{\prime }$.

**Figure 10 entropy-24-00836-f010:**
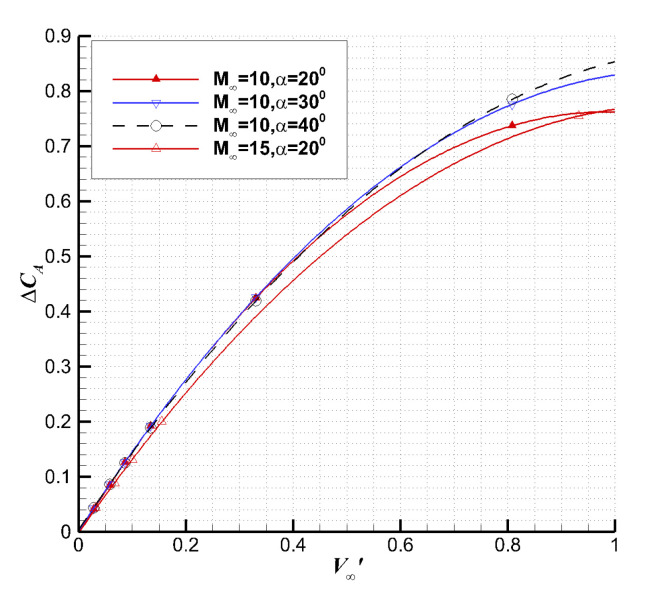
The viscous axial force coefficient vs. $\nu_{\infty }^{\prime }$.

**Figure 11 entropy-24-00836-f011:**
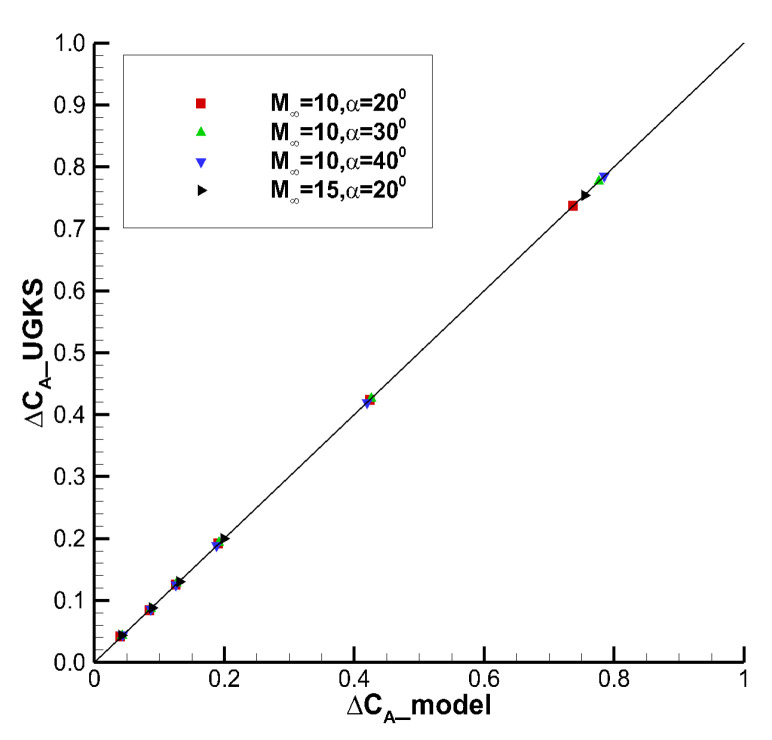
Correlation between viscous interaction model and numerical simulation results.

**Figure 12 entropy-24-00836-f012:**
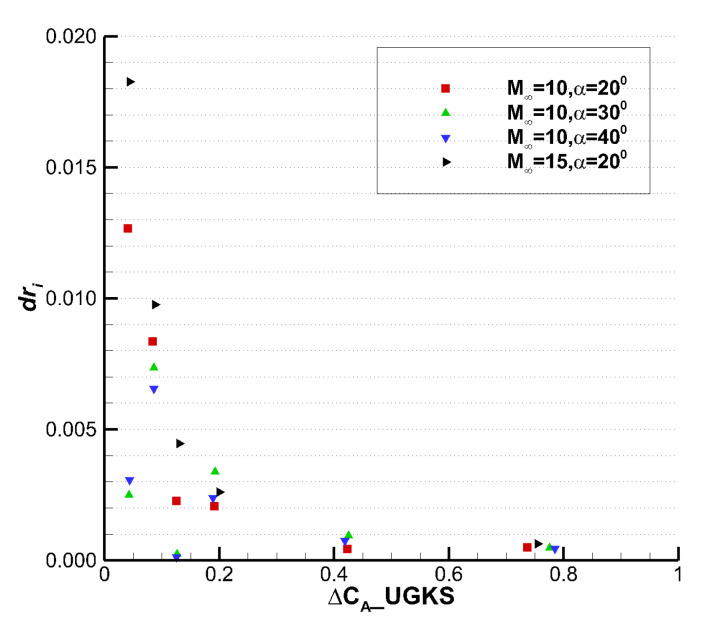
Relative orthogonal distance of the viscous axial force coefficient.

**Table 1 entropy-24-00836-t001:** Free-stream conditions.

Height/km	Mach Number	Angle of Attack/Degrees	Solvers
70, 80, 85, 90, 100, 110	10	20, 30, 40	inviscid, viscous
70, 80, 85, 90, 100, 110	15	20	inviscid, viscous

**Table 2 entropy-24-00836-t002:** Comparison between model predictions and UGKS simulation results.

No	Altitude (km)	UGKS Simulation			Model Prediction			Relative Error		
		dCA	dCN	dCm	dCA	dCN	dCm	dCA	dCN	dCm
1	80	0.0918	0.0448	−0.0221	0.0904	0.0486	−0.0243	−1.58%	8.66%	9.87%
2	90	0.2056	0.1019	−0.0477	0.1993	0.1064	−0.0516	−3.07%	4.47%	8.18%

## Data Availability

The datasets used or analyzed during the current study are available from the corresponding author on reasonable request.

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
