# Peer review of "Nonlinear Modeling Study of Aerodynamic Characteristics of an X38-like Vehicle at Strong Viscous Interaction Regions"

_entropy, 2022, doi:10.3390/e24060836_

Round 1

Reviewer 1 Report

The paper presents a 3D study of high-speed flow over a re-entry geometry based on the kinetic equation solution. Such studies are quite rare and therefore the work is potentially interesting. Unfortunately, the presentation is flawed in many respects and must be improved.

The M=10 is not sufficiently hypersonic. The title of the paper would be justified if the authors usefree stream velocity closer to M=25.

The calculations for 70 and 80 km altitude can be conducted by a NS solver. The realistic simulation should include into consideration the air chemistry and run for air, not monatomic gas. I suggest that the authors remove 70 and 80 km.

Line 31: a reference to a Ph.D. thesis of one of the authors in Chinese is complete inadequate here. Please delete it and provide a meaningful reference to a Godunov solver with an LU-SGS implicit scheme

Line 106: Reference to the original paper of Shakhov is missing. The description of the model is complete.

Line 115: UGKS is a conventional discrete velocity scheme. What does this integral equation have to do with it? When does the velocity grid comes here in integral equartion?

The authors use the S-model equation to simulate a high-speed flow. There have been important recent developments in this area, complete missed by them. First of all, the accuracy of the model equations, including BGK and S-model, was recently evaluated as applied to hypersonic flows (M=10..25) in [1] This reference provides accuracy study for both aerodynamics and heat transfer across various Kn numbers and must be mentioned.

The numerical scheme used appears to be very inefficient due to the use of cartesian velocity mesh. Indeed, the authors need 350 000 cells for M=10.  An adaptive velocity mesh used in [1-2] allows to run with 10 times coarse mesh and up to M=25.  This gives a factor of 10 in efficiency just for the velocity mesh. Apart from that, UGKS is explicit scheme, which is quite inefficient and is probably another factor of 10 slower. How can the authors comment on this?  

In [2] the Smodel solution was compared to DSMC for a complicated Shuttle-like geometry. A short revision of the current state of numerics is given in [3]. For M=25 a complete solution can be computed in less than 10000 core hours. This is state of the art.

Apart from [1-3], there is another approach based on octree-velocity mesh, which is also much more efficient [4]. All these must be included into literature review.

Line 135: what is the mesh resolution near the surface for each altitude? A cut view of the mesh is also needed to visuality evaluate the accuracy. Dimension of the model needs to be specified as well. What is the estimate of accuracy of the computed results? Was influence of mesh resolution studied? If so, provide the results.

Line  165: what is y=500 mm, to what does it correspond? There are no dimensions anywhere in the geometry views.

Line 173: definition of the non-dimensional pressure is missing.

Line 197: not got but computed. Lines 201 - 205 – avoid repeating the same words. The same for lines 257-260.

Figure 12: change the limits in the vertical axis

  1. A. Titarev. Application of model kinetic equations to hypersonic rarefied gas flows // Computers and Fluids. 2018. V. 169. P. 62-70.
  2. A. Titarev, A.A. Frolova, V.A. Rykov, P.V. Vashchenkov, A.A. Shevyrin, Ye.A. Bondar. Comparison of the Shakhov kinetic equation and DSMC method as applied to space vehicle aerothermodynamics // Journal of Computational and Applied Mathematics. 2020. V. 364. P. 1-12. DOI: 10.1016/j.cam.2019.112354.
  3. A. Titarev. Application of the Nesvetay Code for Solving Three-Dimensional High-Altitude Aerodynamics Problems // Computational Mathematics and Mathematical Physics. 2020. Vol. 60, No. 4, pp. 737–748.
  4. Baranger and J. Claudel and N. Herouard and L. Mieussens. Locally refined discrete velocity grids for stationary rarified flow simulations // J. Comput. Phys., 2014, V. 257, P. 572 – 593.

Round 2

Reviewer 1 Report

I am very satisfied with the revision by the authors and happy to recommend the paper for publication.

Reviewer 2 Report

The authors prepared a revised manuscript, fully aligned to my comments and suggestions.